# Small reorganization energy acceptors enable low energy losses in non-fullerene organic solar cells

Yanan Shi[1,2,5], Yilin Chang[1,2,5], Kun Lu [1,2✉], Zhihao Chen[3], Jianqi Zhang [1], Yangjun Yan[1], Dingding Qiu[1,2], Yanan Liu[1], Muhammad Abdullah Adil [1], Wei Ma [4], Xiaotao Hao [3✉], Lingyun Zhu [1✉] & Zhixiang Wei [1,2✉]

Minimizing energy loss is of critical importance in the pursuit of attaining high-performance organic solar cells. Interestingly, reorganization energy plays a crucial role in photoelectric conversion processes. However, the understanding of the relationship between reorganization energy and energy losses has rarely been studied. Here, two acceptors, Qx-1 and Qx-2, were developed. The reorganization energies of these two acceptors during photoelectric conversion processes are substantially smaller than the conventional Y6 acceptor, which is beneficial for improving the exciton lifetime and diffusion length, promoting charge transport, and reducing the energy loss originating from exciton dissociation and non-radiative recombination. So, a high efficiency of 18.2% with high open circuit voltage above 0.93 V in the PM6:Qx-2 blend, accompanies a significantly reduced energy loss of 0.48 eV. This work underlines the importance of the reorganization energy in achieving small energy losses and paves a way to obtain high-performance organic solar cells.

[1] Chinese Academy of Sciences (CAS) key laboratory of nanosystem and hierarchical fabrication, National Center for Nanoscience and Technology, Beijing 100190, China. [2] University of Chinese Academy of Sciences, Beijing 100049, China. [3] School of Physics, State Key Laboratory of Crystal Materials, Shandong University, Jinan, Shandong 250100, China. [4] State Key Laboratory for Mechanical Behavior of Materials, Xi'an Jiaotong University, Xi'an 710049, China. [5] These authors contributed equally: Yanan Shi, Yilin Chang. ✉email: lvk@nanoctr.cn; haoxt@sdu.edu.cn; zhuly@nanoctr.cn; weizx@nanoctr.cn

In recent years, organic solar cells (OSCs) have gained wide attention on account of being lightweight, solution-processable, and ensuring low-cost large-area and semitransparent flexible thin-film device preparation[1–5]. Owing to great strives in material design and device optimization[6–9], the power conversion efficiency (PCE) of OSCs based on Y6 nonfullerene acceptors (NFAs) have reached 18% for the binary devices and 20% for the ternary and tandem devices[10–13]. Nonetheless, compared with inorganic and perovskite solar cells, the OSCs photovoltaic performance still lags behind due to the relatively more significant energy losses[14–17]. Currently, the open circuit voltage ($V_{OC}$) of most high-performance OSCs is still limited to 0.8–0.9 V[18,19], and the energy loss is generally greater than 0.5 eV[20–22]. Thus, to further improve the OSCs efficiencies, it is necessary to gain insight into the source of the energy loss and then to reduce it further through rational molecular design.

Generally, the energy loss in the OSCs mainly arises from two aspects: the driving force for exciton dissociation and non-radiative recombination. Accordingly, many strategies have been proposed to reduce the energy losses; either by reducing the exciton binding energy by regulating molecular packing, which will minimize the driving force required for exciton dissociation[23–25], or, by suppressing the non-radiative recombination during the photoelectric conversion processes, such as exciton decay, charge-transfer (CT) state decay and nongeminate (bimolecular) recombination[22,26–28]. These non-radiative recombinations are all associated with the electron-vibration coupling (namely reorganization energy $\lambda$, which describes the deformations of the molecular geometries in the course of the electron-transfer process and reflects the interactions between electrons and intramolecular vibrations). Moreover, according to the classical Marcus electron-transfer theory ($k_{ET} = V_{if}^2 \sqrt{\frac{\pi}{\lambda k_B T \hbar^2}} \exp\left[-\frac{(\triangle G + \lambda)^2}{4\lambda k_B T}\right]$, where $\lambda$ is the reorganization energy, $V$ represents the electronic coupling between the initial state and the final state, $\Delta G$ is the free energy change)[29], small reorganization energy facilitate reducing the driving force required for exciton dissociation. Therefore, the reorganization energy plays a crucial role in the photoelectric conversion and the energy loss processes for OSCs.

In this work, based on the Y-type NFA as the molecular backbone, by replacing benzothiadiazole (BTZ)-fused core with quinoxaline (Qx)-fused core, two small-molecule acceptors, Qx-1 and Qx-2, have been designed and synthesized (Fig. 1a, and their full names are provided in Supplementary). Historically, Qx and its derivatives have demonstrated various advantages, such as weak electron-deficient properties, rigid plane structures, ease of chemical modification, and multiple substitution positions, which can well regulate their physical and chemical properties[30–35]. Our calculation and experimental results revealed that the reorganization energies during the photoelectric conversion processes of these two acceptors are substantially smaller than the Y6 acceptor, which is beneficial for improving exciton lifetime and diffusion length, promoting charge transport, and suppressing charge recombination. Consequently, significantly reduced energy losses of 0.508 eV and 0.482 eV for Qx-1 and Qx-2 systems, respectively, have been attained, effectively enabling the $V_{OC}$ of both blends to reach over 0.9 V with PM6 (a polymer donor is shown in Supplementary Fig. 1) as a donor. Accordingly, a high PCE of 18.2% in the PM6:Qx-2 blend is obtained with a high $V_{OC}$ of 0.934 V, $J_{SC}$ of 26.5 mA cm$^{-2}$, and fill factor (FF) of 73.7%. To the best of our knowledge, the obtained energy loss is the smallest for the binary OSCs with PCEs over 17% reported to date. This work, therefore, underlines the importance of the reorganization energy on achieving small energy loss in organic active materials and paves a way to obtain high-performance OSCs.

## Results

**Design, optoelectronic characterization, and reorganization energy.** Based on the Y-type NFA as the molecular backbone, by fusing Qx derivatives into the heteroaromatic cores, two acceptors named Qx-1 and Qx-2 respectively have been synthesized, and their chemical structures are shown in Fig. 1a. The detailed synthesis processes and purification methods are listed in Supplementary discussion and Supplementary Fig. 2. The solubility of Qx-1, Qx-2, and Y6 are 36.2 mg/ml, 13.9 mg/ml, and 28.2 mg/ml in chloroform, respectively (see Supplementary Fig. 3 and Supplementary Table 1), indicating that these molecules have good solubility and can be dissolved in common solvents, which is beneficial for device fabrication.

The energy levels of Qx-1 and Qx-2 have been estimated from the electrochemical cyclic voltammetry (CV) method, and the results are shown in Fig. 1b[36,37]. In our investigated A–DA′D–A type Qx-1 and Qx-2 acceptors, we use a weaker electron-withdrawing Qx core to replace BTZ core of Y6 acceptor, which would weaken the intramolecular charge transfer (ICT) effect and cause an increased bandgap. In addition, the enhanced electron-donating ability of the Qx core leads to an increase in both the highest occupied molecular orbital (HOMO) and the lowest unoccupied molecular orbital (LUMO) energies of Qx-1 and Qx-2. The higher LUMO energy level facilitates a higher $V_{OC}$ in the devices[38]. Furthermore, both the energy offsets between the HOMOs of donor and acceptor for hole transfer and the LUMOs of donor and acceptor for electron transfer are largely decreased, which further reduces the voltage loss of exciton dissociation. The normalized UV-vis absorption spectra and photoluminescence (PL) spectra of Y6, Qx-1, and Qx-2 in the solution and neat films have been shown in Fig. 1c-d (detailed optical and electrochemical properties of these acceptors are summarized in Supplementary Fig. 4, Supplementary Table 2, and Supplementary Table 3). Since Qx-1 and Qx-2 molecules have a similar molecular backbone, their absorption data has no significant difference in dilute chloroform solution. Moving from solution to the solid films, their absorption spectra have all been red-shifted, suggesting strong intermolecular $\pi$-$\pi$ interactions[39]. In both solutions and films, the stoke′s shifts from Y6 to Qx-1 and Qx-2 are decreased, suggesting smaller excited-state relaxations in Qx-1 and Qx-2, which is beneficial for the associated voltage losses. This is in agreement with the reduced reorganization energy for the transition between the ground state and the first excited state ($S_1$). In addition, compared with the solutions, the films exhibit relatively larger stoke′s shifts, especially for Y6. This is presumably due to the fact that there exists an energy disorder for the $S_1$ state in the films, and the excitons on the molecules with higher $S_1$ energy can transfer to the molecules with lower $S_1$ energy to emit photons.

Moreover, to further reveal the molecular packing patterns in the solid state, single crystals of Qx-1 and Qx-2 by solvent diffusion method have been grown and the results are shown in Fig. 1e, f (also see Supplementary Figs. 5 and 6, and Supplementary Table 4). Apparently, both single crystals present one-dimensional arrangements via $\pi$-$\pi$ stacking, which differ from Y6 single crystal with a three-dimensional $\pi$–$\pi$ stacking between adjacent molecules[25,40–43]. However, in the solid film, the molecular packing is more isotropic, and it is also possible to form a good three-dimensional packing structure. Furthermore, we carried out atomistic molecular dynamic (MD) simulations, and the results are shown in Supplementary Figs. 7 and 8. For both Qx-1 and Qx-2, there are a variety of $\pi$-$\pi$ stacking modes with different degrees of spatial overlaps, including not only between the end groups but also between the cores, for example, C, Y, W, and S-type. Moreover, ca. 490 and 560 $\pi$-$\pi$ stacking dimers are found in the simulated film containing 400 molecules.

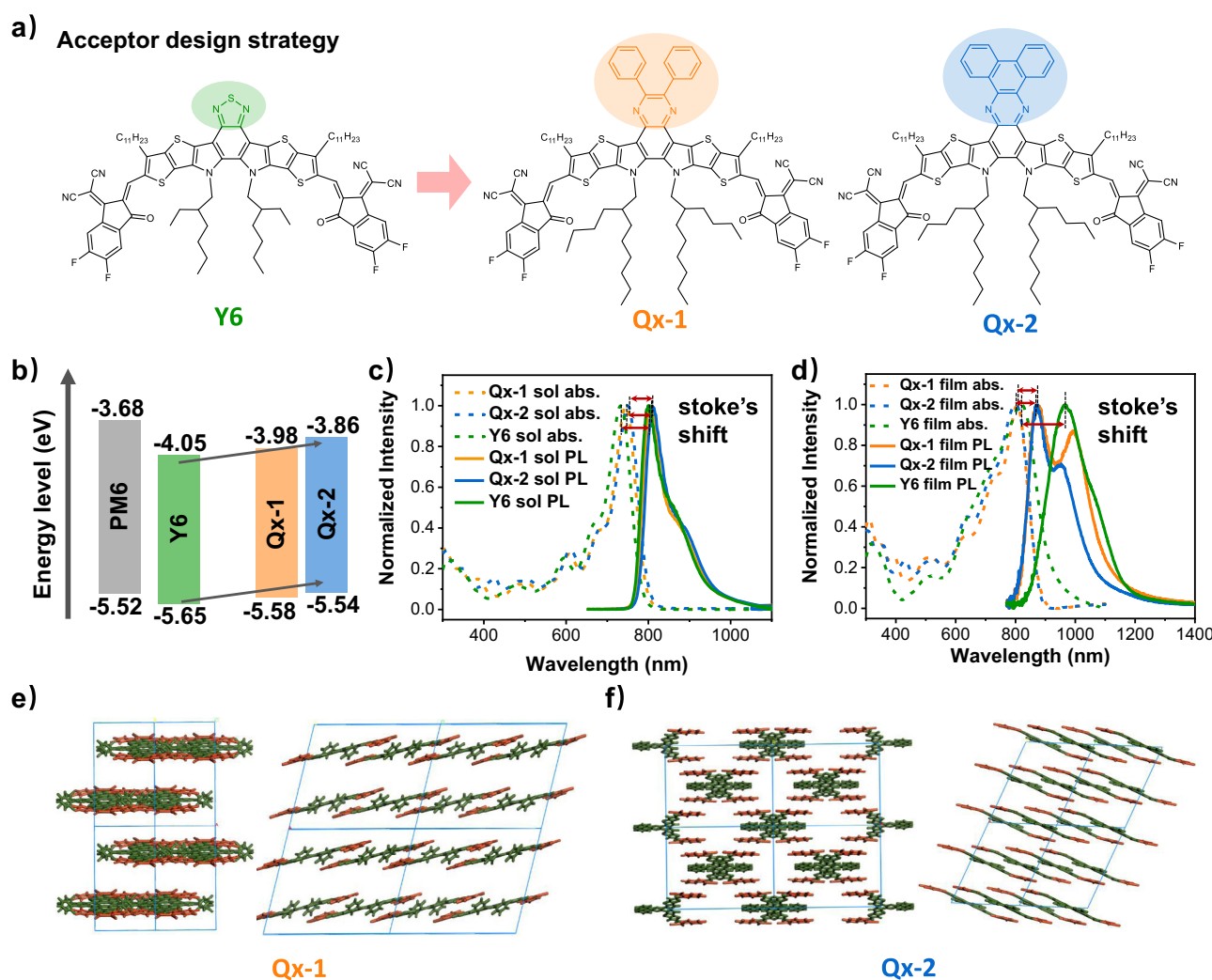

**Fig. 1 Molecular design and optoelectronic characterization of Qx-1 and Qx-2. a** Chemical structures of Y6, Qx-1 and Qx-2 acceptors. **b** Energy level diagram of the related materials. **c** UV–vis absorption (abs.) and photoluminescence (PL) spectra of acceptors in the solution. **d** UV–vis absorption (abs.) and photoluminescence (PL) spectra of acceptors in the films. **e**, **f** The molecular packing patterns of Qx-1 and Qx-2 in the single-crystal structure.

Such high π-π stacking ratios help to increase the molecular interactions and efficient charge transport.

To further understand the effect of Qx moieties on the electronic structure properties of acceptors, the reorganization energies between different electronic states during photoelectric conversions have been calculated at the tuned-ωB97XD/6-31G (d, p) level of theory. As shown in Fig. 2a and Supplementary Table 5 in Supplementary, the reorganization energy for the $S_0 \rightarrow S_1$ transition ($\lambda_{S0 \rightarrow S1}$) is related to the geometry relaxation in the $S_1$ state after light absorption, and the reorganization energy for the $S_1 \rightarrow S_0$ transition ($\lambda_{S1 \rightarrow S0}$) is associated with non-radiative exciton decay from the $S_1$ excitons to the ground state ($S_0$). Their sum corresponds to reorganization energy for exciton diffusion ($\lambda_{EET}$, excitation energy transfer). There are two pathways for the exciton dissociation to CT state process: the dissociation of donor excitons via electron transfer and the dissociation of acceptor excitons via hole transfer. They correspond to the reorganization energy for the $S_0 \rightarrow$ anion ($\lambda_{S0 \rightarrow anion}$) and $S_1 \rightarrow$ anion transitions ($\lambda_{S1 \rightarrow anion}$), respectively. Both of them are related to reorganization energy for exciton dissociation ($\lambda_{ED}$). In the case of the non-radiative charge recombination ($\lambda_{CR}$), it is associated with the reorganization energy for the anion $\rightarrow S_0$ transition ($\lambda_{anion \rightarrow S0}$). The sum of $\lambda_{S0 \rightarrow anion}$ and $\lambda_{anion \rightarrow S0}$ ultimately corresponds to the reorganization energy for electron transport.

According to Marcus theory, the smaller the reorganization energy, the smaller the driving force needed for exciton dissociation, and consequently, the faster exciton transfer and charge transport rate. As shown in Fig. 2b, compared with Y6, replacing BTZ-fused moiety with Qx-fused moiety leads to a dramatic decrease of around 0.02−0.04 eV in the reorganization energy between the $S_0$ and $S_1$ transitions. These results indicate that the exciton decay has been suppressed, and thus, the Qx-fused moiety helps promote exciton transfer. Moreover, the corresponding calculation results are also in good agreement with the smaller stoke's shifts observed in the experimental measurements for the Qx-1 and Qx-2 systems. Similarly, during the process from exciton dissociation to CT state, it is noted that because of similar geometries of the $S_1$ and anion states, the reorganization energy for the exciton dissociation via hole transfer ($\lambda_{S1 \rightarrow anion}$) turned out to be very small for all the acceptors. This is again helpful in reducing the voltage loss in the exciton dissociation process. Moreover, the reorganization energy has also been significantly reduced for the exciton dissociation via electron transfer ($\lambda_{S0 \rightarrow anion}$), which is beneficial for obtaining efficient charge generation under a low driving force. From the CT state to charge-separated (CS) state, the CT state decays to the ground state, or some separated free charge carriers recombine back to the ground state. Both processes generally lead to energy

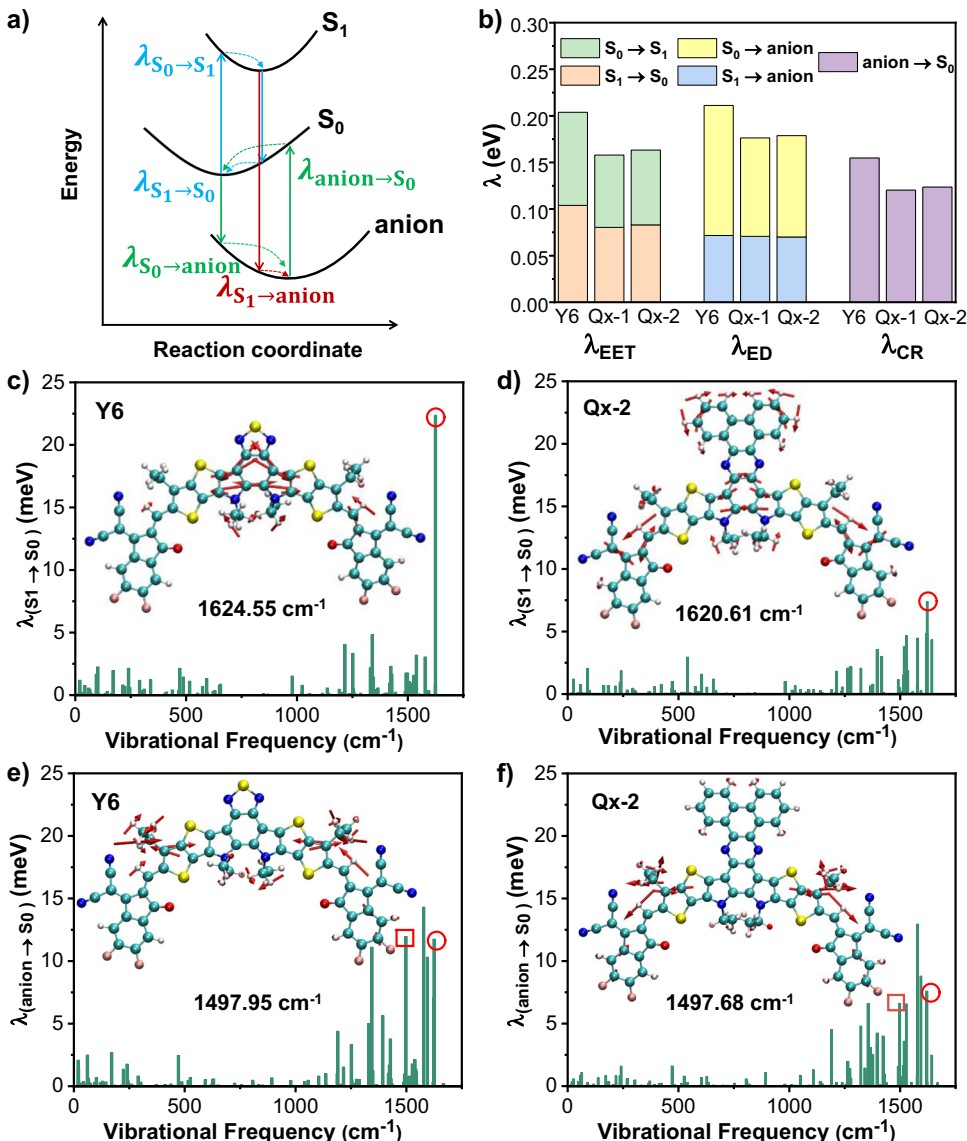

**Fig. 2 Reorganization energy of Y6, Qx-1, and Qx-2 acceptors. a** Illustration of the related transitions among the ground state (S₁), the lowest singlet excited state (S₁), and the anionic state during the photoelectric conversion processes, taking the acceptor as an example. **b** The calculated reorganization energies for the mutual transitions between the different electronic states at the level of ωB97XD/6-31G(d,p) with tuned ω values in Y6, Qx-1, and Qx-2 acceptors. **c–f** Contributions of each vibrational mode to the reorganization energy for the S₁ → S₀ and anion → S₀ transitions of Y6 and Qx-2. Illustration of the displacement vectors for the vibrational normal modes marked by circles (at around 1625 cm⁻¹) and squares (at around 1498 cm⁻¹) are inserted. The length of displacement vectors stands for the magnitude of vibrational strength.

losses. Interestingly, the current results reveal the reorganization energies for the transitions between the anion and S₀ states are much smaller for Qx-1 and Qx-2 with respect to Y6, which is beneficial to reduce non-radiative recombination loss and promote electron transport.

The reorganization energy can further be decomposed into contributions from each vibrational mode in the acceptors, and the results are shown in Fig. 2c–f (the results of Qx-1 are shown in Supplementary Fig. 9). For Y6, there exists one dominated vibrational mode that contributes to the reorganization energy from S₁ to S₀ transition, which corresponds to the stretching of C–C bonds of the central heteroaromatic moiety at a high frequency of 1624 cm⁻¹. Interestingly, after fusing Qx derivatives into the heteroaromatic cores, this vibration is dramatically inhibited and thus facilitates the reduction of exciton decay and promotes exciton transfer. Similarly, in the case of transition from anion to S₀ state, there are four dominated vibrational modes,

which include stretching of C–C bonds accompanied by C–H bonds bending at a high frequency above 1500 cm⁻¹ (See Supplementary Fig. 10). These vibrations are significantly suppressed in Qx-1 and Qx-2, resulting in reduced non-radiative recombination and energy loss. Ultimately, these results highlight the importance of reducing the energy loss to improve the efficiency of OSCs by suppressing the C-C bonds stretching.

**Photovoltaic performance and energy loss**. The photovoltaic performance of Qx-1 and Qx-2 have been analyzed by fabricating the conventional structure with PM6 as the donor (Fig. 3a). All detailed optimizing processes are shown in Supplementary Tables 6 and 7. The statistical distributions of the best PCEs (*ca.* 30 pieces), density–voltage (*J–V*) characteristics, and the external quantum efficiency (EQE) curves of the photovoltaic devices have been provided in Fig. 3b–d. The photovoltaic performance parameters are listed in Table 1. Accordingly, by comparing the

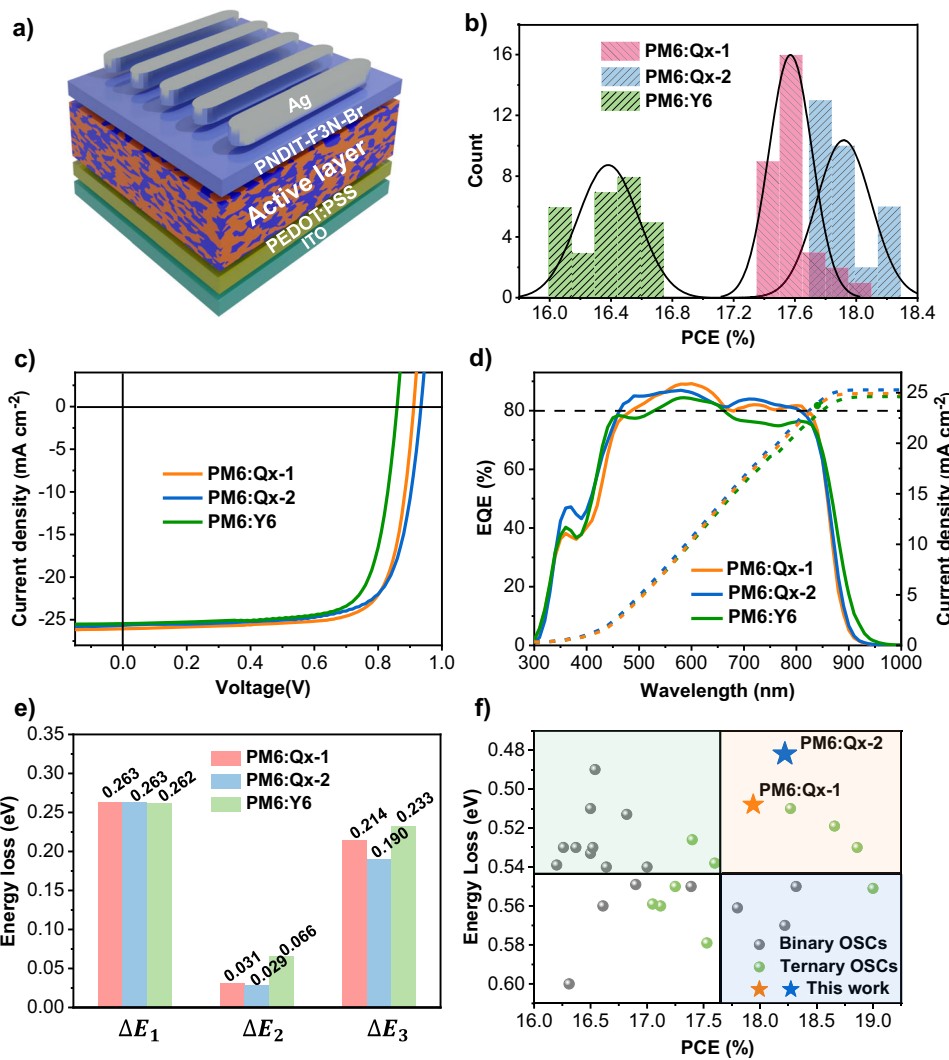

**Fig. 3 Photovoltaic performance and energy loss of Qx-1, Qx-2, and Y6 with PM6 as the donor. a** The conventional device architecture. **b** Statistics distribution of best PCE for *ca*.30 pieces. **c** The density–voltage (*J–V*) curves. **d** The external quantum efficiency (EQE) spectra and the integrated current densities from the EQE spectra of the optimal device. **e** Statistical diagram of energy loss. **f** Plots of the PCE against energy loss for various systems.

**Table 1 Photovoltaic parameters of OSCs with the donor PM6.**

| Acceptors | $V_{OC}$ (V) | $J_{SC}$ (mA cm$^{-2}$) | $J_{SC}$ [a] (mA cm$^{-2}$) | FF (%) | PCE (%) |
|---|---|---|---|---|---|
| Qx-1 | 0.911 (0.912 ± 0.004) | 26.1 (25.6 ± 0.3) | 24.9 | 75.5 (75.4 ± 0.7) | 17.9 (17.6 ± 0.1) |
| Qx-2 | 0.934 (0.935 ± 0.004) | 26.5 (26.0 ± 0.5) | 25.3 | 73.7 (73.9 ± 1.2) | 18.2 (17.9 ± 0.2) |
| Y6 | 0.859 (0.848 ± 0.010) | 25.6 (25.8 ± 0.3) | 24.6 | 75.3 (74.6 ± 1.2) | 16.6 (16.4 ± 0.2) |

The average parameters were obtained from 30 independent devices.
The error bars correspond to the standard deviation of 30 independent devices.
[a]The calculated $J_{SC}$ values from the EQE curves.

device performance of the acceptors Qx-1 and Qx-2 with the PM6:Y6 system, a significant improvement in the Qx-based devices can be observed, as both the Qx-1 and Qx-2 systems demonstrated very high $V_{OC}$ values over 0.9 V. Consequently, the device based on Qx-2 has been able to show an extraordinary PCE of 18.2% with a high $V_{OC}$ of 0.934 V, $J_{SC}$ of 26.5 mA cm$^{-2}$, and FF of 73.7%. For devices based on Qx-1, on the other hand, the best PCE of 17.9% with a $V_{OC}$ of 0.911 V, $J_{SC}$ of 26.1 mA cm$^{-2}$, and FF of 75.5% has been attained. To the best of our knowledge, the $V_{OC}$ obtained for Qx-2 devices is the highest for the OSCs with PCE higher than 17% to date (previous work summary in Supplementary Tables 8 and 9). Furthermore, the

integrated current densities from the EQE curves match well with the integrated current density values from the *J–V* tests (within the 5% error range), indicating the proposed performance parameters have good accuracy and reliability. The certified efficiencies are provided in Supplementary Figs. 11 and 12, and Supplementary Table 10. In terms of device stability, the Qx-based devices demonstrated good storage stability in the glove box at room temperature (See Supplementary Fig. 13), as both acceptors can still attain PCEs of 16%, with almost similar $V_{OC}$ and $J_{SC}$ values after being stored for 960 h.

To probe the charge transport behaviors for the blend films, hole ($\mu_h$) and electron ($\mu_e$) mobilities have been measured by

**Table 2 Energy loss of the devices based on Qx-1, Qx-2 and Y6 acceptors with PM6 donor.**

| Devices | $E_g^{pv a}$ (eV) | $EQE_{EL}$ | $qV_{OC}^{SQ}$ (eV) | $qV_{OC}^{rad}$ (eV) | $E_{loss}$ (eV) | $\Delta E_1$ (eV) | $\Delta E_2$ (eV) | $\Delta E_3$ (eV) | $V_{OC}^{Cal}$ (V) |
|---|---|---|---|---|---|---|---|---|---|
| PM6:Qx−1 | 1.420 | $2.53 \times 10^{-4}$ | 1.157 | 1.120 | 0.508 | 0.263 | 0.031 | 0.214 | 0.912 |
| PM6:Qx−2 | 1.422 | $6.60 \times 10^{-4}$ | 1.159 | 1.130 | 0.482 | 0.263 | 0.029 | 0.190 | 0.941 |
| PM6:Y6 | 1.419 | $1.21 \times 10^{-4}$ | 1.157 | 1.091 | 0.561 | 0.262 | 0.066 | 0.233 | 0.858 |

$^a$The optical bandgap ($E_g^{pv}$) was determined from the derivatives of the EQE curve and the mean peak energy (calculated by the Supplementary equation 13).

using the space charge limited current (SCLC) method (See Supplementary Fig. 14 and Supplementary Table 11)[44]. The $\mu_h$ and $\mu_e$ of PM6:Qx-1 blend film came out to be $3.64 \times 10^{-4}$ and $1.88 \times 10^{-4}$, respectively, while the PM6:Qx-2 blend film revealed a $\mu_h$ and $\mu_e$ of $3.39 \times 10^{-4}$ and $2.13 \times 10^{-4}$, respectively. Compared with PM6:Y6, relatively balanced device mobility has been attained for the acceptors Qx-1 and Qx-2, and the $\mu_h/\mu_e$ is 1.94 and 1.59 for Qx-1 and Qx-2, respectively, which is beneficial for the charge transport. Similarly, the effect of charge recombination behavior of these devices has also been explored by measuring the $J$–$V$ curves at different light intensities[45,46]. Consequently, the trap-assisted recombination appears to be suppressed in both Qx-1 and Qx-2-based OSCs (See Supplementary Fig. 15a, b). Likewise, to evaluate the exciton dissociation and charge extraction properties, the dependence of the photocurrent density ($J_{ph}$) on the effective voltage ($V_{eff}$) of these devices has also been investigated (See Supplementary Fig. 15c)[47]. Under short-circuit conditions, the exciton dissociation efficiency for the devices based on the Qx-1 and Qx-2 blend turns out to be 97.6% and 97.0%, respectively. Such high exciton dissociation and charge extraction values help in explaining the high $J_{SC}$ and, ultimately, high PCE from both systems. Notably, as mentioned above, Qx-2 has a higher π-π stacking ratio in the films, which helps to increase intermolecular interactions and efficient electron transport. From Supplementary Fig. 14 and Supplementary Table 11, our experimental measurements also show that the electron mobility of both pure and mixed Qx-2 film is higher than that of Qx-1, which favors the Qx-2 system with a thicker active layer and obtaining higher $J_{SC}$ (105 nm for Qx-1 and 128 nm for Qx-2). Moreover, the more balanced hole and electron mobility of Qx-2 facilitates improving FF parameters. However, the $J_{SC}$ and FF of the devices often restrict each other. Therefore, a lower FF parameter is obtained in Qx-2.

To further gain insight into the high $V_{OC}$ obtained in Qx-based systems, the energy losses ($E_{loss}$) in both devices have been analyzed. According to the Shockley-Queisser (SQ) limit, the $E_{loss}$ in OSCs can be divided into three parts:[17] $E_{loss} = \Delta E_1 + \Delta E_2 + \Delta E_3$, and the corresponding results have been summarized in Fig. 3e and Table 2. The $\Delta E_1$ is the inevitable energy loss that depends on the $E_g^{pv}$ of the absorber for a definite solar spectrum and temperature ($\Delta E_1 = E_g - qV_{OC}^{SQ}$), $V_{OC}^{SQ}$ is the output voltage in the SQ limit model. Here, the $\Delta E_1$ value of both Qx-1 and Qx-2 blend films are 0.263 eV, the same as the Y6 blend. The $\Delta E_2$ refers to the radiative recombination loss below the bandgap and is attributed to non-step function absorption ($\Delta E_2 = qV_{OC}^{SQ} - qV_{OC}^{rad} = qV_{OC}^{rad}$)[48,49]. Because of the presence of CT state in the OSCs, whose energy is usually lower than the optical gap, there is a partial energy loss, which corresponds to the driving force required for exciton dissociation[50]. The measured electroluminescence (EL) spectra of the PM6:Qx-1 and PM6:Qx-2 blend films are very similar to the corresponding neat acceptors without additional emission peaks from the CT states, while the PM6:Y6 blend shows a slightly larger red-shift as compared to the neat acceptor (See Supplementary Fig. 16). This phenomenon indicates that the energy offsets for exciton dissociation in the PM6:Qx-1 and PM6:Qx-2 blends have been

largely reduced. As a result, the $\Delta E_2$ of PM6:Qx-1 and PM6:Qx-2 blend films are around two-fold lower than the PM6:Y6 blend, as small as 0.031 eV and 0.029 eV, respectively. These experimental results are in excellent agreement with the calculations, showing that the reorganization energy for the exciton dissociation in Qx-1 and Qx-2 systems has significantly been reduced, facilitating efficient charge generation at low driving forces.

Similarly, $\Delta E_3$ is the non-radiative recombination ($\Delta E_3 = qV_{OC}^{nonrad} = -k_B T \ln(EQE_{EL})$), where $EQE_{EL}$ is the radiative quantum efficiency of the device when charge carriers are injected in dark conditions. This part of energy loss is ascribed from the non-radiative decay of the CT states to the ground state and the recombination of the separated free charges. As discussed above, when the energy offset between donor and acceptor is small, it will increase the hybridization of the CT state with the highly emissive local excited state, thereby reducing the non-radiative energy loss. Moreover, our calculations show that the reorganization energies for the transitions between the anion and $S_0$ states are much smaller than for Qx-1 and Qx-2 with respect to Y6, which is beneficial for reducing recombination loss. Furthermore, photoluminescence quantum yield (PLQY) is of critical importance to suppress the non-radiative voltage loss. Here, the measured PLQYs of Qx-2 and Qx-1 are 12.34% and 7.37%, respectively, much higher than Y6 (6.42%, see Supplementary Table 12), which inhibits the non-radiative recombination loss. As a result, the PM6:Qx-1 and PM6:Qx-2 blends exhibit a high $EQE_{EL}$ of $2.53 \times 10^{-4}$ and $6.60 \times 10^{-4}$ (See Supplementary Fig. 17), resulting in a small loss of 0.214 eV and 0.190 eV due to the reduced non-radiative recombination, respectively. Since both the driving force required for exciton dissociation and non-radiative recombination have now been significantly reduced, the energy losses in both PM6: Qx-1 and PM6: Qx-2 blend is dramatically decreased, as small as 0.508 eV and 0.482 eV, respectively. Remarkably, the PM6:Qx-2 blend has the lowest energy losses reported to date for the binary OSCs with PCEs over 17% (Fig. 3f).

**Morphology, exciton, and charge dynamics.** In addition to rational molecular design, a good phase morphology with moderate phase separation size is also one of the indispensable factors for obtaining high-efficiency OSCs. Here, grazing-incidence wide-angle X-ray scattering (GIWAXS), atomic force microscopy (AFM), transmission electron microscopy (TEM), and resonant soft X-ray scattering (R-SoXS) were used for morphology characterization of the neat and blend films of two acceptors. In the neat films, Qx-1 exhibits stronger crystallization than Qx-2 (Fig. 4a–c), which is consistent with a higher packing density of the single crystal structure of Qx-1 than that of Qx-2. However, in the blend film, Qx-2 exhibits a better ability to maintain crystallization than Qx-1(Fig. 4d–f). By comparing the corresponding crystal plane parameter between single crystal and GIWAXS of neat films, we can understand the main stacking pattern of Qx-1 and Qx-2 (See Supplementary Figs. 18 and 19). Compared with the Qx-1 neat films, the stacking mode of (200) as the plane unfavorable charge transfer disappears in the blend films.

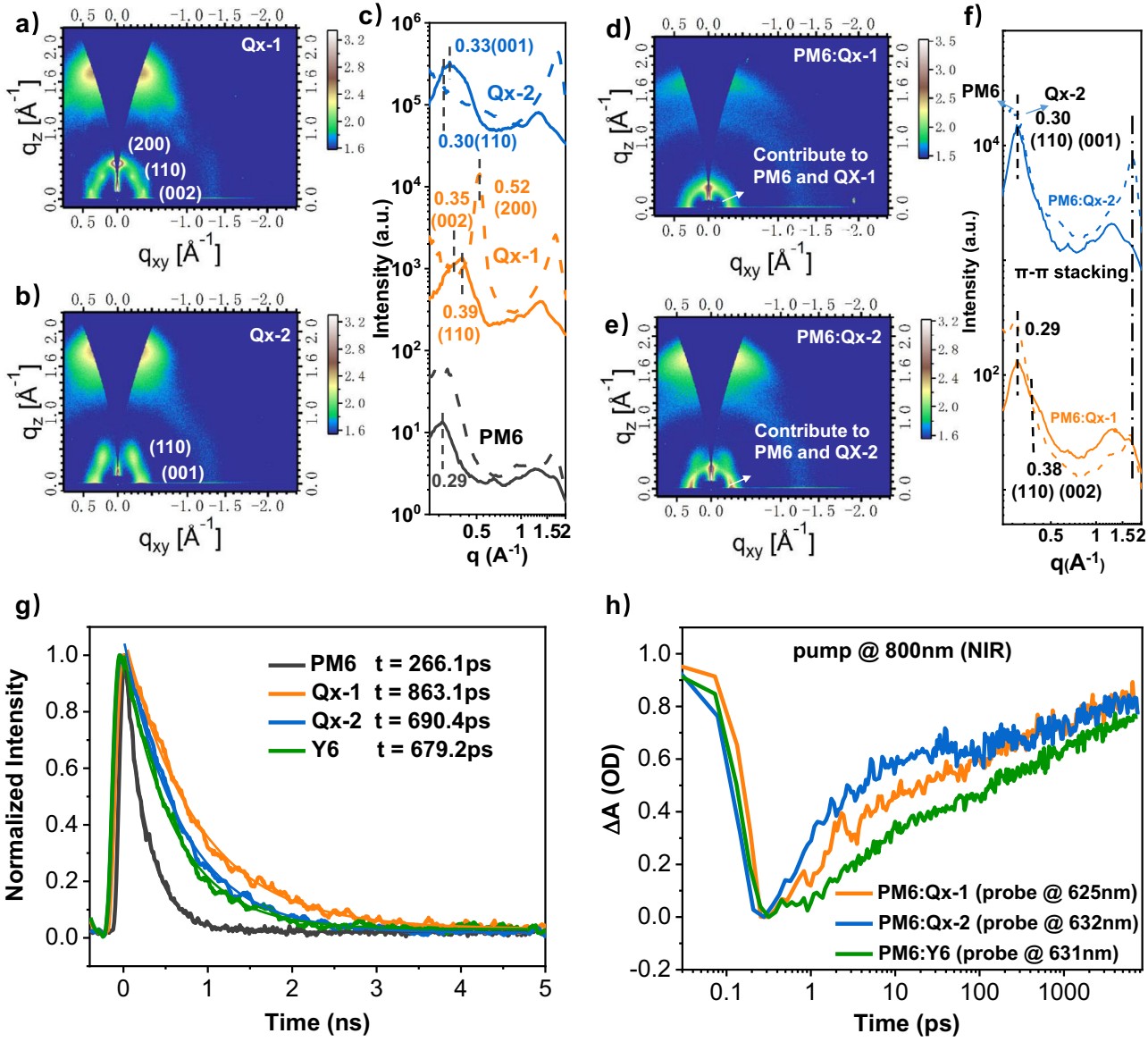

**Fig. 4 Morphology, exciton and charge dynamics. a–c** Two-dimensional GIWAXS patterns and in-plane (solid lines) and out-of-plane (dashed lines) cuts of the neat films; **d–f** Two-dimensional GIWAXS patterns and in-plane (solid lines) and out-of-plane (dashed lines) cuts of blend films; **g** Time-resolution photoluminescence (TRPL) spectrum of donor and acceptors in neat films; **h** Transient kinetic traces of PM6 ground state bleaching (GSB) probing at around 630 nm for the blend films.

However, the stacking mode with (002) or (001) and (110) as the plane exist in both the Qx-1 and Qx-2 neat and blend films. These stacking modes indicate the backbone orientation on the substrate is almost vertical, which is conducive to charge transport to the electrodes. Overall, all of the blend films exhibit face-on arrangement and good crystallinity, which facilitate the charge transfer and suppress the charge recombination. Similarly, from the AFM images (See Supplementary Fig. 20), the Qx-1 and Qx-2 blend films exhibit clearly fibrous structure, which favors excellent charge transport and thus, enable the devices to achieve higher $J_{SC}$ and FF[51,52]. Compared with the results in neat films, the fibrous structure in the blend films is mainly derived from the aggregation characteristics of PM6. Furthermore, the Qx-1 appears to have a more non-planar structure and a poor ability to maintain crystallization, resulting in higher roughness (1.66 nm). On the contrary, the Qx-2 has a planar structure and a better ability to maintain crystallization, resulting in a smaller roughness (0.898 nm). For TEM images of the optimal blend films (See Supplementary Fig. 20), the planar and rigid structure of Qx-2 acceptors is more conducive to breaking the strong self-aggregation of PM6 and obtaining a more uniform phase distribution and smaller domain size, which is consistent with the Resonant Soft X-ray Scattering (R-SoXS) results (See Supplementary Fig. 21). The scattering peaks of the Qx-1 and Qx-2 systems are at $q = 0.07$ and $0.15$ nm$^{-1}$, respectively, indicating that the phase domain size of the Qx-2 system (21.63 nm) is smaller than the Qx-1 system (43.56 nm). This phase size is the relatively ideal active layer morphology of OSCs and thus facilitates in achieving higher exciton dissociation and charge transport efficiency.

Apart from film morphology, the investigation of exciton and charge dynamics can give us a deeper understanding of the exciton transport and charge generation behavior in the active layer. Hence, the time-resolved photoluminescence (TRPL) spectrum of neat and blended films has been carried out to determine the exciton lifetime, and the results are shown in

Fig. 4g. Thus, the exciton lifetime of both Qx-1 and Qx-2 neat films (863.1 ps and 690.4 ps, respectively) turned out to be greater than Y6 (679.2 ps). In addition, as discussed above, the quantum yield of Qx-2 and Qx-1 also came out to be higher than Y6. These results point out that Qx-1 and Qx-2 have a lower exciton decay than Y6. Likewise, an obvious fluorescence quenching in the blends as compared to the lifetime of the neat film has also been observed, indicating an efficient exciton dissociation, especially for the PM6:Qx-2 system (See Supplementary Fig. 22). The exciton-exciton annihilation (EEA) method has been used to investigate the exciton diffusion length $(L_D)$[53,54]. The calculated $L_D$ of Qx-1 and Qx-2 turned out to be comparable to Y6, as values 10.6 nm, 9.5 nm and 12.6 nm for Y6, Qx-1 and Qx-2, respectively (See supplementary Fig. 23 and Supplementary Table 13). These results are also in accordance with the previous ones and indicate that reducing the reorganization energy is an effective pathway to suppress exciton decay and improve exciton transport and lifetime.

The photo-induced CT behavior has further been investigated by measuring the Transient absorption spectrum (TAS) of the neat and blend films (See Supplementary Figs. 24 and 25). A beam of 400 nm and 800 nm excitation light has been used to form PM6 donor excitons and Qx-1 and Qx-2 acceptor excitons, respectively. The negative signals in the range from 600 to 650 nm have been assigned to the ground state bleach (GSB) of PM6. Meanwhile, a GSB peak at 825 nm is observed that matches the absorption spectrum of acceptors. Since the absorption spectra of the donor and acceptors can be well separated, the hole transfer dynamics can be clearly investigated by selectively exciting the acceptor molecules. The pump wavelength of 800 nm is set to excite the acceptors in the blend film. Accordingly, an additional GSB peak has been observed at 630 nm that attributed to the PM6 donor, resulting from the ultrafast hole transfer from Qx-1 and Qx-2 acceptor to the donor at the donor–acceptor interface. In addition, both PM6:Qx-2 and PM6:Qx-1 blends exhibit a much faster hole transfer rate than Y6, which benefits the effective charge generation in the OSCs (Fig. 4h). These results are in good agreement with the high performance of two Qx-based blends.

## Discussion

In this work, two acceptors of Qx-1 and Qx-2 with bigger volume and higher rigid Qx as central core have been synthesized. It has been observed that by suppressing the molecular vibration of stretching of C-C bonds in both acceptors, we find that the reorganization energies during the photoelectric conversions have significantly reduced as compared to the conventional Y6 acceptor. Thus, the energy losses arising from the exciton dissociation and non-radiative recombination have greatly been reduced. Hence, a relationship between the reorganization energy and the energy losses has been well established. Furthermore, the film morphology, exciton, and charge dynamics results prove the Qx-1 and Qx-2 acceptors with smaller reorganization energy have better aggregation patterns, better exciton lifetime and longer diffusion lengths, contributing to the effective charge generation in corresponding OSCs. As a result, the PM6:Qx-1 blend produced a PCE of 17.9% with an energy loss of 0.508 eV, whereas the PM6:Qx-2 system attained a high PCE of 18.2% with an energy loss of 0.482 eV, significantly outperforming its Y6 counterpart. To the best of our knowledge, the obtained energy loss is the smallest for the binary OSCs with PCEs over 17% reported to date. This work underlines the importance of the reorganization energy for achieving small energy loss in organic active materials and thereby provides a strategy toward the high performance of OSCs.

## Methods

**Materials**. The synthetic route of acceptors Qx-1 and Qx-2 is illustrated in Supplementary Fig. 2. Polymer donor PM6 was purchased from Solar Materials. The detailed synthetic procedures of Qx-1 and Qx-2 and the corresponding structural characterizations can be found in the Supplementary.

**Single-crystal growth**. Single crystals of Qx-1 and Qx-2 were grown by the liquid diffusion method at room temperature. An appropriate amount of methanol is transferred to a concentrated chloroform solution, which will form crystals over time. Single crystal diffraction was collected at low temperatures protected by liquid nitrogen in accordance with standard procedures for reducing X-ray radiation damage through the use of the single-crystal X-ray diffractometer (model is XtaLAB PRO 007HF(Mo), manufactured by Rigaku). The X-ray crystallographic coordinates for structures reported of Qx-1 and Qx-2 have been deposited at the Cambridge Crystallographic Data Centre (CCDC), under deposition numbers 2120380–2120381.

**UV-vis absorption**. Measured by Perkin Elmer Lambda 950 spectrophotometer.

**Electrochemical cyclic voltammetry (CV)**. Measured with an electrochemical workstation (VMP3 Biologic, France) with a Pt disk coated with blend film, a Pt plate, and an Ag$^+$/Ag electrode acting as the working, counter, and reference electrodes, respectively, in a 0.1 mol/L tetrabutylammonium phosphorus hexafluoride (Bu$_4$NPF$_6$) acetonitrile solution.

**Device fabrication and measurement**. All the PSC devices were fabricated by using an conventional structure of ITO/PEDOT:PSS/PM6:acceptors/PNDIT-F3N-Br/Ag, where PNDIT-F3N-Br (poly[(9,9-bis(3′-((N,N-dimethyl)-N-ethylammonium)propyl)-2,7 fluorene)-alt-5,5′-bis(2,2′-thiophene)-2,6-naphthalene-1,4,5,8-tetracaboxylic-N,N′-di(2ethylhexyl)imide]dibromide) and PEDOT:PSS (poly(3,4-ethylenedioxythiophene:poly(styrenesulfonate)) were respectively used as electron-transport and hole-transport interlayer. The blended solution was prepared by mixing PM6 and acceptors into chloroform with the addition of a small amount of chloronaphthalene (0.6%, v/v), the mix solution was stirred at 50 °C in chloroform for 1.5 h until they dissolved. The optimal device conditions of Qx-1 and Qx-2 were prepared by mixing donor and acceptors in a 1:1.5 and1:1.3 weight ratio, respectively, into chloroform with the addition of a small amount of 1-chloronaphthalene (0.6%, v/v) and under thermal annealing at 100 °C and 110 °C for 10 min, respectively.

A solar simulator was used for J–V curves measurement under AM 1.5 G (100 mW cm$^{-2}$). Newport Oriel PN 91150 V Sibased solar cell was applied for light intensity calibration. J–V measurement signals were recorded by a Keithley 2400 source-measure unit. The device area of each cell was approximately 4mm$^2$. And the measurements were performed by scanning voltage from −1 to 1 V with a voltage step of 10 mV and delay time of 1 ms. Oriel Newport system (Model 66902) equipped with a standard Si diode was used for EQEs test in air condition.

**Mobility measurements**. The electron mobility was acquired with the device structure of Al/active layer or neat acceptors/PNDIT-F3N-Br/Al, the hole mobility was obtained by preparing the structure of ITO/PEDOT:PSS/active layer/MoOx/Ag. The current density-voltage (J–V) curves in the range of 0–5 V were obtained by a Keithley 2420 Source-Measure Unit in the dark.

**Energy loss**. Highly sensitive EQE was measured using an integrated system (PECT-600, Enlitech), where the photocurrent was amplified and modulated by a lock-in instrument. EQE$_{EL}$ measurements were performed by applying external voltage/current sources through the devices (ELCT-3010, Enlitech). EQE$_{EL}$ measurements were performed for all devices according to the optimal device preparation conditions.

**Morphology characterization**. Transmission electron microscopy (TEM) images were acquired on Tecnai G2 F20 U-TWIN TEM instrument. The atomic force microscopy (AFM) characterization was performed by Bruker Multimode 8 in ScanAsyst Mode in air. Grazing incidence wide angle X-ray scattering (GIWAXs) measurement was conducted at the beamline of 7.3.3 at the Advanced Light Source (ALS). Resonant Soft X-ray Scattering (R-SoXS): R-SoXS transmission measurements were performed at beamline 11.0.1.2 at the Advanced Light Source (ALS).

**Exciton and charge dynamics**. The excitation and emission spectra, the time-resolution photoluminescence (TRPL) spectrum and absolute quantum yield (QY) test by transient/steady state fluorescence spectrometer, manufactured by Edinburgh Instruments, model number is FLS1000. The PL spectrum of film and solution use a xenon lamp to excite the light source, use 750 nm pump for film excitation, and use 600 nm pump for solution excitation. Absolute quantum yield using an integrating sphere and a blank quartz plate as a reference to measure the absolute quantum yield. TA measurements were performed on an Ultrafast Helios

pump-probe system in collaboration with a regenerative amplified laser system from Coherent.

**Reporting summary**. Further information on research design is available in the Nature Research Reporting Summary linked to this article.

## Data availability

Source data are provided with this paper. All data generated or analyzed in this study are included in this published article and its supplementary information. The X-ray crystallographic coordinates for structures generated in this study have been deposited at the Cambridge Crystallographic Data Centre (CCDC), under deposition numbers 2120380–2120381. These data can be obtained free of charge from the Cambridge Crystallographic Data Centre via www.ccdc.cam.ac.uk/data_request/cif.

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

## Acknowledgements

This work was financially supported by National Natural Science Foundation of China under Grant Nos. 22073020 (L.Z.), 21773040 (L.Z.), 21822503 (K.L.), 51973043 (K.L.), 22135001 (Z.W.), 21721002 (Z.W.), the Strategic Priority Research Program of the Chinese Academy of Sciences under Grant No. XDB36000000 (Z.W.), and the CAS-CSIRO Joint Project of Chinese Academy of Sciences under Grant No. 121E32KYSB20190021 (K.L.). We acknowledge Prof. Zhishan Bo and Prof. Xinjun Xu of Beijing Normal University for the help in the device performance test.

## Author contributions

Y.S. synthesized and characterized the NFAs of Qx-1 and Qx-2. Y.C. and Y.Y. performed the device fabrication and characterization. Y.S. tested the energy loss data. L.Z. carried out theoretical calculations. Y.S. and L.Z. grew the single crystals and analyzed the single-crystal structures of Qx-1 and Qx-2. Z.C. and X.H. measured and analyzed transient absorption spectroscopy. J.Z. measured and analyzed the GIWAXS. D.Q. and Y.L. performed the TEM measurements of the single crystal. Y.S. performed the morphology characterization and analyzed the data. W.M. performed the R-SoXS measurements. M.A. polished the language. Z.W., K.L., and L.Z. supervised the project, Y.S., L.Z., K.L., and Z.W. wrote the paper.

## Competing interests

The authors declare no competing interests.
