## [Peer Review File · Nature Communications]

Small reorganization energy acceptors enable low energy losses in non-fullerene organic solar cellsREVIEWER COMMENTS

Reviewer #1 (Remarks to the Author):

In this manuscript, Shi et al. developed two new acceptors of Qx-1 and Qx-2 with quinoxaline (Qx)-containing fused core. A decent PCE over 18% was achieved in the PM6:Qx-2 binary device with a low energy loss of 0.482 V. Device performance, energy loss, exciton and charge dynamics and morphology were fully characterized and analyzed. The result is very interesting, underlining the importance of the reorganization energy for achieving small energy loss in organic active materials. Therefore, I recommend this work to be published in Nature Communication after addressing the following issues.

1. Page 5, line 91, the authors statement that the lowest unoccupied molecular orbital (LUMO) energies of Qx-1 and Qx-2 have been upshifted, resulting in slightly increased bandgaps. I think the description is not right, the bandgap is not directly related to the molecular energy levels, which is mainly affected by the ICT effect and molecular packing. Similarly, page 6, line 105, the authors described that the higher energy level and larger E_{gopt} do favor obtaining a higher V_{oc} , how a large band gap produces a high V_{oc} , Is there a direct connection between the E_{gopt} and V_{oc} ? The authors should revise them carefully.

2. From the view of chemical structures, can the authors explain that the stoke's shift of Qx-1 and Qx-2 is significantly smaller than Y6.

3. In the whole article, the author mainly compares Qx-1 and Qx-2 with Y6, but ignores the comparison of device parameters between Qx-1 and Qx-2. For example, why the Qx-1-based device yields higher FF but lower JSC relative to those of Qx-2-based device, the authors did not give an explanation. The structure-property relationship of Qx-1 and Qx-2 needs to be investigated.

4. Some descriptions are not rigorous, for example, "the obtained energy loss is the smallest for the binary OSCs with PCEs over 16% reported to date" and "the V_{oc} obtained for Qx-2 devices is the highest for the OSCs with PCE higher than 16% to date", a PCE over 16% with V_{oc} of 0.94 V has been achieved (10.1016/j.joule.2020.05.010) and a PCE over 16% with energy loss of 0.48 V has been reported (10.1038/s41566-019-0573-5), the authors should revise these descriptions.

5. Increasing the photoluminescence quantum yield (PLQY) is of critical importance to suppress the nonradiative voltage loss. However, when introducing non-radiative recombination losses, the authors ignored the role of PLQY and only emphasized the role of reorganization energies.

6. If possible, the authors could provide the certified efficiency to confirm the high JSC and PCE.

Below are some grammatical or miscellaneous errors.

7. Page 3, line 58, “by replacing thiophene-fused core with quinoxaline (Qx)-fused core”, can be better expressed as “by replacing benzothiadiazole-fused core with quinoxaline (Qx)-fused core”. Please also check the rest of the manuscript as this term has been used for numerous times in the manuscript.

8. Page 3, line 63, “which can better.....” can be better expressed as “which can well....”.

9. Page 3, line 63, “which is beneficial for improved exciton lifetime and diffusion length, promote charge transport and suppress charge recombination” should be expressed as “which is beneficial for improving exciton lifetime and diffusion length, promoting charge transport and suppressing charge recombination.”

10. Page 5, line 91, “unccupied” should be “unoccupied”.

11. Page 6, line 97, the “Detailed” should strat with a lower case “d”.

12. Page 6, line 112, “to future reveal” can be better expressed as “to further reveal”.

13. References 26 and 29 are repeated.

Reviewer #2 (Remarks to the Author):

Reorganization energy plays an essential role during the photoelectric conversions of organic photovoltaics. In this manuscript, the authors reported two new acceptors, Qx-1 and Qx-2, with smaller reorganization energies than the conventional Y6 acceptor, thus enabling a low energy loss and outstanding power conversion efficiency of 18.2 % in the binary devices. Furthermore, the DFT calculations proved that the reorganization energies during the photoelectric conversions have significantly reduced compared to Y6, leading to greatly reduced energy losses. Therefore, this work presents a joint experimental and theoretical study to establish the relationship between the reorganization energy and energy losses. The finding is indeed inspiring and vital for further developing organic photoactive materials to minimize the energy losses of organic solar cells. However, several problems should be solved before being accepted:

(1) Can the authors provide the data of the UV-vis absorption and photoluminescence spectra of acceptors in the solution to see the Stoke’s shift?

(2) Qx-1 and Qx-2 present one-dimensional arrangements in the single crystals; how about the molecular packing in the solid films, and what is the difference from the single crystals?

(3) The authors state that these molecules show good solubility, the solubility of these acceptors should be provided.

(4) The authors mentioned that “according to the classical Marcus electron-transfer theory, small reorganization energy facilitates reducing the driving force required for exciton dissociation” in the Introduction part. Therefore, it is better to include the Marcus equation to show the role of reorganization energy clearly.

(5) some typos need to be corrected; for example, in Figure 3b, the legend should be PM6:Y6. In P3, line 52, “meolecular” should be molecular.

(6) A recently published paper should be added: Joule 2021, DOI: 10.1016/j.joule.2021.12.017.

Reviewer #1:

In this manuscript, Shi et al. developed two new acceptors of Qx-1 and Qx-2 with quinoxaline (Qx)-containing fused core. A decent PCE over 18% was achieved in the PM6:Qx-2 binary device with a low energy loss of 0.482 V. Device performance, energy loss, exciton and charge dynamics and morphology were fully characterized and analyzed. The result is very interesting, underlining the importance of the reorganization energy for achieving small energy loss in organic active materials. Therefore, I recommend this work to be published in Nature Communication after addressing the following issues.

Response to comment: We are grateful to the Reviewer for the very positive comments on our work.

1. *Page 5, line 91, the authors statement that the lowest unoccupied molecular orbital (LUMO) energies of Qx-1 and Qx-2 have been upshifted, resulting in slightly increased bandgaps. I think the description is not right, the bandgap is not directly related to the molecular energy levels, which is mainly affected by the ICT effect and molecular packing. Similarly, page 6, line 105, the authors described that the higher energy level and larger Egopt do favor obtaining a higher Voc, how a large band gap produces a high Voc, Is there a direct connection between the Egopt and Voc? The authors should revise them carefully.*

Response: Thank you for your valuable suggestions. We agree that the bandgap is affected by the ICT effect. In our investigated A–DA'D–A type Qx-1 and Qx-2 acceptors, we used a weaker electron-withdrawing quinoxaline (Qx) core to replace benzothiadiazole (BTZ) core of Y6 acceptor, which would weaken the ICT effect and cause an increased bandgap. In addition, the enhanced electron-donating ability of Qx core leads to an increase in both the highest occupied molecular orbital (HOMO) and the lowest unoccupied molecular orbital (LUMO) energies of Qx-1 and Qx-2. The higher LUMO energy level of acceptors favor obtaining a higher Voc.

For the optical bandgap, we also agree the Reviewer's point that there is no direct connection between the E_g^{opt} and Voc. For the sake of clarity, we have deleted these discussions on Page 4 in the revised manuscript.

We have modified the bandgap discussions as below:

“In our investigated A–DA'D–A type Qx-1 and Qx-2 acceptors, we use a weaker electron-withdrawing Qx core to replace BTZ core of Y6 acceptor, which would weaken the intramolecular charge transfer (ICT) effect and cause an increased bandgap. In addition, the enhanced electron-donating ability of Qx core leads to an increase in both the highest occupied molecular orbital (HOMO) and the lowest unoccupied molecular orbital (LUMO) energies of Qx-1 and Qx-2. The higher LUMO energy level facilitates a higher V_{oc} in the devices.”

2. *From the view of chemical structures, can the authors explain that the stoke's shift of Qx-1 and Qx-2 is significantly smaller than Y6.*

Response: Thank you for the comment. Only from the view of chemical structures, there is no straightforward relationship between the stoke's shift and chemical structure. Here, compared with Y6, Qx-1 and Qx-2 have “larger” intermediate cores and stronger molecular rigidity, which may be beneficial to reduce the excited-state geometry relaxations, resulting in a smaller stoke's shift. To further understand the intrinsic reasons, we need to carry out DFT calculations for the reorganization energy from S_1 to S_0 transition. Compared with Y6, replacing thiophene-fused moiety with quinoxaline-fused moiety leads to a dramatic decrease of around 0.02 – 0.04 eV in the reorganization energy. The reorganization energy is then decomposed into contributions from each vibrational mode. For Y6, there exists one dominated vibrational mode that contributes to the reorganization energy from S_1 to S_0 transition, which corresponds to the stretching of C–C bonds of the central heteroaromatic moiety at a high frequency of 1624 cm^{-1} . Interestingly, after fusing quinoxaline derivatives into the heteroaromatic cores, this vibration is dramatically inhibited, thus facilitating the reduction of the relaxation of excited state relaxations.

3. *In the whole article, the author mainly compares Qx-1 and Qx-2 with Y6, but ignores the comparison of device parameters between Qx-1 and Qx-2. For example, why the Qx-1-based device yields higher FF but lower JSC relative to those of Qx-2-based device, the authors did not give an explanation. The structure-property relationship of Qx-1 and Qx-2 needs to be investigated.*

Response: Thanks a lot for your valuable suggestion. For both Qx-1 and Qx-2, our atomistic molecular

dynamic (MD) simulations find that there are around 490 and 560 π - π stacking dimers in the simulated film containing 400 molecules, respectively. Such higher π - π stacking ratios in Qx-2 help to increase the molecular interactions and efficient electron transport. As seen from Figure S13 and Table S11, our experimental measurements also show that the electron mobility of both pure and mixed Qx-2 film is higher than that of Qx-1, which favors the Qx-2 system with a thicker active layer and obtaining higher J_{sc} (105 nm for Qx-1 and 128 nm for Qx-2). Moreover, the measured μ_h/μ_e of Qx-1 and Qx-2 is 1.94 and 1.59, respectively, and the more balanced hole and electron mobility of Qx-2 facilitates improving FF parameters. However, the J_{sc} and FF of the devices often restrict each other. Therefore, a lower FF parameter is obtained in Qx-2.

We have added corresponding discussions on Page 13 in the revised manuscript as below:

“Notably, as mentioned above, Qx-2 has a higher π - π stacking ratio in the films, which helps to increase intermolecular interactions and efficient electron transport. From Fig. S13 and Table S11, our experimental measurements also show that the electron mobility of both pure and mixed Qx-2 film is higher than that of Qx-1, which favors the Qx-2 system with a thicker active layer and obtaining higher J_{sc} (105 nm for Qx-1 and 128 nm for Qx-2). Moreover, the more balanced hole and electron mobility of Qx-2 facilitates improving FF parameters. However, the J_{sc} and FF of the devices often restrict each other. Therefore, a lower FF parameter is obtained in Qx-2.”

Table. R1 The hole mobility and electron mobility.

Films	μ_e ($\text{cm}^2 \text{V}^{-1} \text{s}^{-1}$)	μ_h ($\text{cm}^2 \text{V}^{-1} \text{s}^{-1}$)	μ_h/μ_e
PM6:Qx-1	1.88×10^{-4}	3.64×10^{-4}	1.94
PM6:Qx-2	2.13×10^{-4}	3.39×10^{-4}	1.59
Qx-1	4.56×10^{-4}		
Qx-2	6.98×10^{-4}		

4. Some descriptions are not rigorous, for example, “the obtained energy loss is the smallest for the binary OSCs with PCEs over 16% reported to date” and “the V_{oc} obtained for Qx-2 devices is the

highest for the OSCs with PCE higher than 16% to date”, a PCE over 16% with VOC of 0.94 V has been achieved (10.1016/j.joule.2020.05.010) and a PCE over 16% with energy loss of 0.48 V has been reported (10.1038/s41566-019-0573-5), the authors should revise these descriptions.

Response: Thank you very much for your advice. To make it more clearly, we have revised our descriptions as below:

“To the best of our knowledge, the obtained energy loss is the smallest for the binary OSCs with PCEs over 17% reported to date.

To the best of our knowledge, the V_{oc} obtained for Qx-2 devices is the highest for the OSCs with PCE higher than 17% to date (previous work summary in Supplementary Table S8-S9).”

We also added the references mentioned above in Figure 3f. The updated figure is as below:

Figure R1. Plots of the PCE against energy loss for various systems.

5. *Increasing the photoluminescence quantum yield (PLQY) is of critical importance to suppress the nonradiative voltage loss. However, when introducing nonradiative recombination losses, the authors ignored the role of PLQY and only emphasized the role of reorganization energies.*

Response: Thank you for your valuable suggestion. We agree that PLQY is a critical parameter to reflect the nonradiative voltage loss since $PLQY = k_r / (k_r + k_{nr})$ (k_r and k_{nr} is the radiative rate and nonradiative rate, respectively). The lower nonradiative recombination is helpful to increase the PLQY. Here, the measured quantum yields of Qx-2 and Qx-1 are 12.34% and 7.37%, respectively, much

higher than that of Y6 (6.42%), which inhibits the nonradiative recombination loss.

We have added the corresponding discussion on Page 14 in the revised manuscript as below:

“Furthermore, photoluminescence quantum yield (PLQY) is of critical importance to suppress the nonradiative voltage loss. Here, the measured PLQYs of Qx-2 and Qx-1 are 12.34% and 7.37%, respectively, much higher than Y6 (6.42%, see Supplementary Table S12), which inhibits the nonradiative recombination loss.”

6. *If possible, the authors could provide the certified efficiency to confirm the high JSC and PCE.*

Response: Following the Reviewer’s suggestion, we have provided independent certification of PM6:Qx-1 and PM6:Qx-2 from the National Institute of Metrology (NIM), and the results are shown in Figure R2. With an area of 2.558 mm² mask, the device based on PM6:Qx-1 shows PCE of 17.6 % with V_{oc} of 0.880 V, J_{sc} of 0.658 mA (25.72 mA cm⁻²), and FF of 77.6%; the device based on PM6:Qx-2 shows PCE of 17.5 % with V_{oc} of 0.887 V, J_{sc} of 0.654 mA (25.57 mA cm⁻²), and FF of 77.1%. The results are generally consistent with our results. The slight decrease in PCE is probably due to the ambient influence on the devices since our devices were transferred without any encapsulation. To further verify the photovoltaic performance of our devices, we have also tested the device of PM6:Qx-2 both in our lab and Prof. Zhishan Bo’s lab (Beijing Normal University, Beijing 100875, P. R. China, zsbo@bnu.edu.cn). The J - V curves and photovoltaic results are summarized in Figure R3 and Table R2. The PCEs obtained from Prof. Bo’s lab are very close to our results.

The related results have been added in Supplementary Figure S10, Figure S11, and Table S10 in the revision.

(a)

(b)

Figure R2. Certification report by National Institute of Metrology (NIM), China. a) PM6:Qx-1 device; b) PM6:Qx-2 device.

Figure R3. The J - V curves of three devices based on PM6:Qx-2. a) in our lab; b) in Prof. Bo's lab.

Table R2. Photovoltaic parameters of three devices based on PM6:Qx-2 measured both in our lab and in Prof. Zhishan Bo's lab.

Number	condition	Voc (V)	Jsc (mA/ cm ²)	FF (%)	PCE (%)
No. 1	Our lab	0.933	25.53	76.13	18.13
	Prof. Bo's lab	0.930	25.64	75.77	18.07
No. 2	Our lab	0.936	26.09	74.69	18.23
	Prof. Bo's lab	0.934	25.81	74.43	17.94
No. 3	Our lab	0.927	25.99	75.46	18.19
	Prof. Bo's lab	0.927	25.81	75.02	17.94

Below are some grammatical or miscellaneous errors

7. Page 3, line 58, “by replacing thiophene-fused core with quinoxaline (Qx)-fused core”, can be better expressed as “by replacing benzothiadiazole-fused core with quinoxaline (Qx)-fused core”. Please also check the rest of the manuscript as this term has been used for numerous times in the manuscript.

8. Page 3, line 63, “which can better.....” can be better expressed as “which can well....”

9. Page 3, line 63, “which is beneficial for improved exciton lifetime and diffusion length, promote charge transport and suppress charge recombination” should be expressed as “which is beneficial for

improving exciton lifetime and diffusion length, promoting charge transport and suppressing charge recombination.”

10. Page 5, line 91, “unccupied” should be “unoccupied”.

11. Page 6, line 97, the “Detailed” should strat with a lower case “d”.

12. Page 6, line 112, “to future reveal” can be better expressed as “to further reveal”.

13: References 26 and 29 are repeated.

Response: Many thanks for the Reviewer’s careful reading. We have carefully checked all the mentioned parts and revised these mistakes carefully.

Reviewer #2 :

Reorganization energy plays an essential role during the photoelectric conversions of organic photovoltaics. In this manuscript, the authors reported two new acceptors, Qx-1 and Qx-2, with smaller reorganization energies than the conventional Y6 acceptor, thus enabling a low energy loss and outstanding power conversion efficiency of 18.2 % in the binary devices. Furthermore, the DFT calculations proved that the reorganization energies during the photoelectric conversions have significantly reduced compared to Y6, leading to greatly reduced energy losses. Therefore, this work presents a joint experimental and theoretical study to establish the relationship between the reorganization energy and energy losses. The finding is indeed inspiring and vital for further developing organic photoactive materials to minimize the energy losses of organic solar cells. However, several problems should be solved before being accepted:

Response to comment: We are grateful to the Reviewer for the very positive comments on our work.

1. *Can the authors provide the data of the UV-vis absorption and photoluminescence spectra of acceptors in the solution to see the Stoke’s shift?*

Response: Thanks a lot for your good suggestion. We have measured the UV-vis absorption and photoluminescence spectra of acceptors in the solution, and the results are added in Figure 1c and Supplementary Table S3.

We have added more discussion on Page 5 of the revised manuscript as below:

In both solutions and films, the stoke's shifts from Y6 to Qx-1 and Qx-2 are decreased, suggesting smaller excited-state relaxations in Qx-1 and Qx-2, which is beneficial for the associated voltage losses. This is in agreement with the reduced reorganization energy for the transition between the ground state and the S_1 state. In addition, compared with the solutions, the films exhibit relatively larger stoke's shifts, especially for Y6. This is presumably due to the fact that there exists an energy disorder for the S_1 state in the films, and the excitons on the molecules with higher S_1 energy can transfer to the molecules with lower S_1 energy to emit photons.

Figure R4. UV-vis absorption and photoluminescence (PL) spectra of acceptors (a) in the solution; (b) in the films.

Table. R3 Stoke's shift of Qx-1, Qx-2 and Y6 in solution and neat films (in the unit of nm).

System	Solution			Film		
	Y6	Qx-1	Qx-2	Y6	Qx-1	Qx-2
Stoke's shift	66	60	58	148	65	68

2. *Qx-1 and Qx-2 present one-dimensional arrangements in the single crystals; how about the molecular packing in the solid films, and what is the difference from the single crystals?*

Response: This is a good question. We have carried out atomistic molecular dynamic (MD) simulations to shed some light on the thin-film molecular packing structures. The computational details

and results are added in Supplementary Figures S6-S7. We have added more discussion on Page 5 of the revised manuscript as below:

“Furthermore, we carried out atomistic molecular dynamic (MD) simulations, and the results are shown in Supplementary Fig. S6 and S7. For both Qx-1 and Qx-2, there are a variety of π - π stacking modes with different degrees of spatial overlaps, including not only between the end groups but also between the cores, for example, C, Y, W, and S-type. Moreover, ca. 490 and 560 π - π stacking dimers are found in the simulated film containing 400 molecules. Such high π - π stacking ratios help to increase the molecular interactions and efficient charge transport.”

Figure R5. (a) Representative morphology of the simulated Qx-1 thin film. (b) Statistics of the possible dimers in thin film by MD simulations. The molecular pair with ≥ 8 interacting atoms (inter-atomic distance is smaller than the sum of the of the atomic van der Waals radii) is regarded as π - π stacking. (c) Representative π - π stacking dimers in the Qx-1 film.

Figure R6. (a) Representative morphology of the simulated Qx-2 thin film. (b) Statistics of the possible dimers in thin film by MD simulations. The molecular pair with ≥ 8 interacting atoms is regarded as π - π stacking. (c) Representative π - π stacking dimers in the Qx-2 film.

3. *The authors state that these molecules show good solubility, the solubility of these acceptors should be provided.*

Response: We have provided the solubility of Qx-1, Qx-2, and Y6 through the test of the UV-vis absorption spectra. First, we obtained the linear relationship between concentration and UV-vis absorption intensity by measuring the UV-vis absorption spectra of chloroform solutions with different known concentrations, and further measured the UV-vis absorption spectra of the diluted saturated solution, and deduced the solubility of the material in chloroform. The details and results are added in Supplementary Figure S2 and Table S1, and added the corresponding discussion on Page 4 of the revised manuscript as below:

“The solubility of Qx-1, Qx-2 and Y6 are 36.2 mg/ml, 13.9 mg/ml and 28.2mg/ml in chloroform, respectively (Supplementary Fig. S2 and Table S1), indicating that these molecules have good

solubility and can be dissolved in common solvents, which is beneficial for device fabrication.”

Figure R5. (a, e, h) UV-vis absorption spectra of acceptors of chloroform solutions with different known concentrations; (b, f, i) the linear relationship between concentration and UV-vis absorption intensity; (c, g, j) the UV-vis absorption spectra of the diluted saturated solution.

Table. R3 UV-vis absorption intensity with different concentrations (in mg/ml).

Qx-1		Standard curve					DSS (*8000)	
Concentration	0.0125	0.0083	0.00625	0.00417	0.0025	0.00125	36.2	
Absorption intensity	1.77	1.202	0.884	0.589	0.372	0.185	0.65	
Qx-2		Standard curve					DSS (*2400)	
Concentration	0.0115	0.00767	0.00575	0.00383	0.0023	0.00115	13.9	
Absorption intensity	1.492	1.045	0.746	0.506	0.285	0.146	0.756	
Y6		Standard curve					DSS (*3500)	

Concentration	0.012	0.008	0.006	0.004	0.0024	0.0012	28.2
Absorption intensity	1.393	0.946	0.712	0.465	0.288	0.137	0.943

*DSS= Diluted saturated solution

4. *The authors mentioned that “according to the classical Marcus electron-transfer theory, small reorganization energy facilitates reducing the driving force required for exciton dissociation” in the Introduction part. Therefore, it is better to include the Marcus equation to show the role of reorganization energy clearly.*

Response: According to the Reviewer’s advice, we have added Marcus equation in the revised manuscript. The modifications are as below:

“Moreover, according to the classical Marcus electron-transfer theory ($k_{ET} = V_{if}^2 \sqrt{\frac{\pi}{\lambda k_B T \hbar^2}} \exp\left[-\frac{(\Delta G + \lambda)^2}{4\lambda k_B T}\right]$, where λ is the reorganization energy, V represents the electronic coupling between the initial state and the final state, ΔG is the free energy change), small reorganization energy facilitate reducing the driving force required for exciton dissociation.”

5. *Some typos need to be corrected; for example, in Figure 3b, the legend should be PM6:Y6. In P3, line 52, “melecular” should be molecular.*

Response: According to the Reviewer’s advice, we have checked the mentioned parts, and corrections are made accordingly.

6. *A recently published paper should be added: Joule 2021, DOI: 10.1016/j.joule.2021.12.017*

Response: Thanks for the Reviewer’s comments. We have added this reference to the introduction part (ref.13).

REVIEWERS' COMMENTS

Reviewer #1 (Remarks to the Author):

The revised version has well addressed the concerns raised by the referees, and I recommend its acceptance in the current form.

Reviewer #2 (Remarks to the Author):

Underlining the essential role of the reorganization energy for achieving small energy loss in organic photovoltaic materials is very important. In this manuscript, the authors' finding is vital for further designing organic photoactive materials to minimize the energy losses of OSCs. The manuscript has been carefully revised according to the comments. Therefore, I recommend that this work be considered for publication in Nature Communication.